# Materialists perceive their high socioeconomic status as justice: Associations with increased political participation

Zhirui Zhao[1], Qi Zhao[1], Su Tao[1]*, Wenchong Du[2]*

**1** School of Marxism, China University of Geosciences (Beijing), Beijing, China, **2** NTU Psychology, School of Social Science, Nottingham Trent University, Nottingham, United Kingdom

* taosu@cugb.edu.cn (ST); vivienne.du@ntu.ac.uk (WD)

## Abstract

Existing research has not reached a consensus on the relationship between subjective socioeconomic status (SSS) and political participation. Thus, this study investigated the psychological variables influencing the relationship between SSS and political participation. Specifically, it explored the mediating role of perceived social justice and the moderating role of materialism value. A sample of 1306 college students was conducted with the MacArthur Scale of SSS, the System Justification Scale, the Material Value Scale, and the Political Participation Behavior Scale. The results showed that: (1) There was a significant positive correlation between SSS, perceived social justice and political participation; materialism value shows a significant negative correlation with SSS, perceived social justice, and political participation. (2) Perceived social justice played a mediating role in the effect of SSS on political participation. (3) Materialism value moderated the relationship between SSS and perceived social justice. Under high materialism value, perceived social justice increases with SSS; however, under low materialism value, this effect is no longer significant. The study enriches our understanding of the underlying psychological mechanisms and marginal effects in the relationship between SSS and political participation.

## Introduction

Political participation behavior is believed to be influenced by socioeconomic status [1]; however, the relationship between the two remains a subject of debate. It is found that individuals with higher socioeconomic status exhibit a more positive attitude toward the country's political system and are more inclined to engage in various political activities, as opposed to those with lower socioeconomic status, who are found to be less likely to participate in political involvement [1,2]. However, theories concerning group identity and group dynamics imply different outcomes. For instance, out group favor ability bias assumes that members of low-status groups often show

**Data availability statement:** All relevant data are within the paper and its Supporting information files.

**Funding:** This research was funded by Humanities and Social Science Projects of the Ministry of Education of China (19YJC710066). The funders provided financial support but had no role in study design, data collection and analysis, decision to publish, or preparation of the manuscript.

**Competing interests:** The authors have declared that no competing interests exist.

a preference for external groups [3]. According to the permeability of group boundaries theory, individuals may seek to leave their current socioeconomic class and join higher classes [4]. Thus, individuals in low socioeconomic status groups often desire to enter higher socioeconomic status groups and may engage in group political participation to change the group's status or advocate for its interests.

The controversy may stem from various studies focusing on different aspects of political participation behaviors [5–7]. However, the more crucial factor could be the divergent internal psychological processes experienced by individuals of different socioeconomic statuses. Individual differences may also lead to distinct perceptions of socioeconomic status, consequently resulting in varied behavioral outcomes.

This study aims to investigate the psychological processes through which an individual's subject socioeconomic status influences political participation, as well as to identify the individual differences that may exert marginal effects in this context.

### Political participation and subjective socioeconomic-status (SSS)

Political participation is an "activity by private citizens designed to influence government decision-making" [8]. Political participation takes various forms, including not only traditional forms such as voting, campaigning, and protesting, but also other forms such as reporting social issues to the media, discussing social issues with others, and participating in social movements or other political groups [9]. Here, we define political participation as a subset of broad political acts that tends policy and government, such as voting, expressing own opinion in an online or realistic world, and working on a political campaign [8].

Socioeconomic status (SES) is a measure of a person's position in the social and economic hierarchy, which composes of objective capital (objective SES) and subjective perception of these capitals (subjective socioeconomic status, SSS) [10]. According to the ideology hypothesis [11], the differences of economic status will affect individual's view of wealth and social justice, and further may affect his/ her political behavior. For instance, Jazmin's research has found that SSS can influence both political behavior and ideology [1]. SSS may represent an individual's subjective cognition of personal and economic data (e.g., education, income), and compared with solitary objective indicators (objective SES), it may be a more accurate predictor of thoughts and behaviors, and show adequate stability over time [12]. Thus, we examine the relationship between SSS and political participation rather than objective SES.

Some studies have indicated a negative relationship between SSS and political participation or social change willingness [13–16]. For example, the social competition strategy suggests that the state of competition will occur when people feel injustice [16]. Individuals in the low SSS group may perceive social division as unreasonable, promoting them to directly compete with members outside the group and engage in more social change activities to pursue greater opportunities and benefits. Additionally, empirical studies revealed that individuals in the low SSS group tend to engage in political participation in forms such as resistance and reform [17].

However, more evidence suggests a positive correlation between SSS and political participation [1,9,11,16]. Firstly, according to System Justification Theory, individuals tend to justify, rationalize, and support existing social, economic, and political systems, even those that disadvantage groups [18]. This inclination is motivated by a desire to reduce cognitive dissonance and maintain stability [19]. As a result, individuals with lower SSS may be less disposed to advocate for change through political participation. Secondly, according to the Resource Model, individuals with lower SSS are less politically engaged compared to those with higher SSS due to a lack of financial resources, such as free time, civic skills, or level of participation necessary for political involvement [20]. For example, the political participation of individuals with lower SSS is not only limited by objective barriers such as not enough time or stressful work, but also by psychological barriers [1,21]. Specifically, social norms related to one's financial resources can be reflected by political participation. Lower SSS individuals may live in areas that don't require political participation such as voting and campaigning as the social norm [22]. Consequently, lower SSS may unwilling to make efforts in the political arena or completely out of this arena because they lack basic political skills or even view political participation as conflicting with their own values [20]. Thirdly, lower SSS may reduce one's trust in the government [23], and lack of social justice which in turn may lead to political inaction.

The results indicating a negative relationship between SSS and political participation primarily point to forms of resistance or disruptive political engagement [17,24,25]. In contrast, studies suggesting a positive relationship tend to focus on more moderate, non-confrontational forms of political participation and offer a more thorough analysis of the psychological processes underlying the relationship between SSS and political participation. In the current stable political environment in China, political participation among youth is generally more moderate [26]. Based on these considerations, we present **Hypothesis 1: SSS is positively correlated with political participation.**

## Perceived social justice, SSS, and political participation

The relationship between SSS and political participation may be indirectly affected by other variables. The "Freedom model of Political Participation in Development" proposes that factors affecting individuals' political participation include not only their level of economic development, but also their perception of social justice [27]. In other words, if a person's SSS is higher, but his perception of social justice is lower, it will also reduce their enthusiasm for political participation. Therefore, the research on political participation behavior should not only focus on individuals' perception of the total amount of development (SSS), but also focus on individuals' perception of the distribution of development achievements (perceived social justice).

Perceived social justice is an individual's perception of the degree of social justice. That is, people make "fair or unfair" judgments on whether the society meets the "due social conditions" as the standard [19]. "Structural status determinism" holds that an individual's perception of distributive justice depends on how much benefit he gets from the distribution behavior. If an individual benefits from distribution, he is more inclined to believe that distribution is fair [2]. Therefore, the perception of social justice often appears with an individual's SSS. In other words, an individual's perception of social justice is determined by SSS. The higher the SSS, the more inclined to think that the whole society is fair, and the more willing to participate in political activities to maintain the current social "equality" system [28,29]. At the same time, existing studies have found that SSS is closely related to individuals' perceptions of social justice [9,16,23]. For example, some scholars have found a certain correlation between SSS and perceived social justice after analyzing the data of CSS2006, CSS2008, CSS2013, and CSS2015 in the past ten years. Those who consider themselves to be in the middle and above SSS have slightly higher social justice perceptions than those who identify themselves as in the lower middle SSS [30].

In addition, the research finds that perceived social justice is an important variable to explain political participation, that is, perceived social justice is an important driving force for individuals to have a trusting experience of society and the trusting experience ultimately prompts individuals to participate in political behaviors [16,23]. Moreover, the current researches have pointed out that there is a significant positive correlation between perceived social justice and political

participation behavior [9,23]. Based on this, we present **Hypothesis 2: Perceived social justice plays a mediating role in the influence of SSS on political participation.**

### Materialism value, SSS, and perceived social justice

The relationship between SSS and perceived social justice may be influenced by individual differences, and the role of SSS may vary depending on individuals' perspectives on money and social status, referred to as materialism value [31]. Materialism is generally defined as "individual differences in people's endorsement of values, goals, and associated beliefs that center on the importance of acquiring money and possessions that convey status" [32]. Numerous studies have shown that materialism value significantly impacts individuals' perceived social justice [33–35]. Liu's research found that individuals with a high level of materialism value will value material gains more and may be more sensitive to inequality in the distribution of residents, whereas non-materialists are less sensitive [32]. Sirgy's Spillover Theory posits that materialists often set unrealistic goals for living standards that are too high for them to achieve [36]. As a result, they experience more dissatisfaction with their standard of living compared to non-materialists. Consequently, this dissatisfaction spills over into other areas of life, resulting in a general dissatisfaction with life and a perception of societal unfairness.

Therefore, this study assumes that materialism value may indirectly affect perceived social justice by moderating the relationship between SSS and perceived social justice. Specifically, the presence and extent of materialistic values could alter the strength or direction of this relationship. Consequently, we propose **Hypothesis 3: Materialism value can moderate the relationship between SSS and perceived social justice.** Specifically, individual who espouse materialistic values are more likely to experience dissatisfaction when assessing their SSS, potentially perceiving greater societal unfairness. This perception might lead to apathy towards societal issues and a reduced willingness to engage in political activities. The hypothetical model of this study is shown in Fig 1.

## Method

### Ethical statement

The study protocol was reviewed and granted ethical approval by the Academic Committee of the School of Marxism, China University of Geosciences (Beijing). Before filling out the questionnaire, participants were given detailed information about the study, including its purpose, potential risks, and benefits. Throughout the process, participants could withdraw from the study at any time. Submitting the questionnaire online was considered as providing informed consent. All data were collected, sorted, and analyzed anonymously, without any identifiable personal information.

### Inclusivity in global research

Additional information regarding the ethical, cultural, and scientific considerations specific to inclusivity in global research is included in the Supporting Information (S3 Checklist).

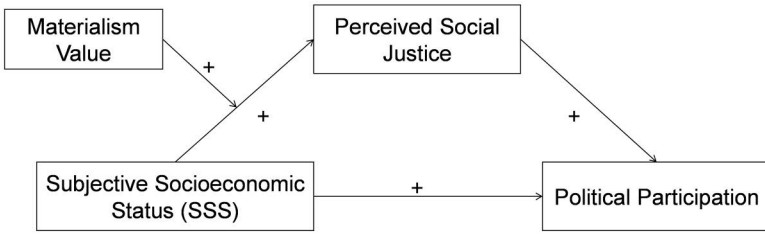

**Fig 1. The proposed moderated mediation model.**

## Participants

Data were collected via Tencent Questionnaire (https://wj.qq.com), a widely used platform in China. The participants were recruited through two methods: first, data were collected via cluster sampling from courses, with students receiving course points as a reward for completing the survey; second, a snowball sampling technique [37] was employed, where students who had completed the survey invited their peers from different universities to participate. This approach ensured that the sample included participants from both top-tier and regular universities, as well as from a diverse range of regions and institutions, thereby enhancing the representativeness of the sample. The homogeneity of university students allowed for more effective control of confounding variables, which strengthened the examination of relationships among various variables in the model being tested [38].

The sample consisted of 1,350 university students from China. Invalid questionnaires with too short response time or missing items were excluded. Out of the distributed questionnaires, 1,306 valid responses were returned, yielding a rate of 96.74%. The participants were 726 males (55.6%), 577 females (44.2%), and 3 individuals did not report their sex. The mean age of the participants was 20.46 years (SD = 1.21). The grade distribution included 743 freshmen and sophomores (56.9%), 454 juniors and seniors (34.8%), and 107 graduate students (8.2%), with two participants not reporting their grade. Participants were from 16 of China's 34 provinces, autonomous regions, and municipalities, with 881 (67.5%) from urban areas, 422 (32.3%) from rural areas, and 3 not reporting their home location.

## Measures

**SSS.** This study utilized the MacArthur Scale of SSS to measure participants' subjective SES [12]. Participants were instructed to indicate their perceived position in society on a ladder ranging from 1 to 10. The bottom of the ladder (step 1) represents individuals who are worst off (lowest wealth and income, least education, and worst jobs). The top of the ladder (step 10) represents individuals who are the best off (highest wealth and income, highest education, and best jobs). Higher scores indicate a higher level of subjective social status. Scores are treated as continuous variables, and the scale's validity and reliability have been well-documented in previous research [12].

**Perceived social justice.** This study utilized 8 items from the System Justification Scale to measure perceived social justice [19]. The items were as follows: 'In general, you find society to be fair'. Participants rated each item on a 5-point scale (1 = strongly disagree to 5 = strongly agree). Average scores were computed for all the items, with items 2 and 6 being reversely scored. Higher scores indicate higher levels of perceived social justice. In this study, Cronbach's α was 0.81, which is greater than 0.7, indicating that the internal consistency reliability is acceptable [39].

**Materialism value.** This study utilized 18 items from the Material Values Scale compiled by Richins and Dawson [31] (1992) and revised by Li and Guo (2009) to measure materialism, including three underlying factors (i.e., material centrality, material happiness, and material success) [31,40]. Participants were asked to rate the extent of agreement on a 5-point scale (1 = strongly disagree, 5 = strongly agree). Higher scores indicate a higher level of materialism value. In this study, Cronbach's α was 0.80, which is greater than 0.7, indicating that the internal consistency reliability is acceptable [39].

**Political participation.** In this study, we utilized a self-developed scale to measure political participation behavior. The questionnaire was developed following a strict development process, outlined in Supporting Information (S1 Scale Development in S1 File). It consists of 10 items, including actions such as 'visiting the official website of the government, relevant departmental websites, or government-affiliated new media' 'sending comments to the school through the Principal's mailbox' 'joining a political club at school'. Participants were asked to rate the frequency of each of the 10 items on a 5-point scale (1 = never, 2 = rarely, 3 = sometimes, 4 = often, 5 = always). Higher scores indicate more political participation. We employed Confirmatory Factor Analysis (CFA) to validate the factor structure, which involved 102 participants. Results of the CFA showed: $\chi^2/df = 1.80$, $CFI = 0.96$, $TLI = 0.93$, and $RMSEA = 0.09$. According to the criteria, $\chi^2/df < 3$, $CFI$ and $TLI \geq 0.90$, and $RMSEA$ between 0.08 and 0.10 indicate a good fit [41]. These results indicate that the model fits the data well. Regarding reliability, Cronbach's α coefficient for internal consistency reliability was 0.93, which is greater than 0.9, indicates excellent internal consistency [39].

## Data analysis

Preliminary analyses, including Harman's single-factor test, descriptive statistics, and Pearson correlation analysis, were conducted using IBM SPSS Statistics 22. Subsequently, the moderated mediation model was tested using the PROCESS macro for SPSS. Bootstrapping with 95% confidence intervals was employed to assess the statistical significance of the moderated mediation model, based on 5000 random samples [42]. Additionally, we conducted a Structural Equation Model (SEM) analysis using AMOS 24.0 to validate the validity of the moderated mediation model.

## Results

### Common method biases

Using Harman's single-factor test, non-rotating exploratory factor analysis was conducted on all measurement items. The results showed that 6 common factors with eigenvalues greater than 1, and the first common factor explained 23.65% of the total variance, which was less than the critical threshold of 40%. This indicates that there are no obvious common method biases in this study.

### Descriptive analyses and correlations

The mean, standard deviation, and correlation coefficients of this study are presented in Table 1. SSS has a significant positive correlation with political participation and perceived social justice. The independent variable, dependent variable, and proposed mediating variable are all significantly correlated, indicating the feasibility of conducting a mediation analysis. Additionally, materialism value shows a significant negative correlation with the SSS, perceived social justice, and political participation.

### The influence of SSS on political participation and the mediating of perceived social justice

Firstly, PROCESS 4.1 Model 4 was used to test the significance of the mediating effect with SSS as the independent variable, perceived social justice as the mediating variable, and political participation as the dependent variable. Gender and age were included as the control variables.

The results showed that SSS had a significant positive prediction on political participation ($\beta = 0.09$, $t = 3.33$, $p < 0.001$), thereby supporting Hypothesis 1. After including perceived social justice in the regression model, the association between SSS and political participation remained significant ($\beta = 0.07$, $t = 2.73$ $p < 0.01$), with the 95% confidence interval of the direct effect of SSS on political participation being [0.03, 0.14]. Furthermore, SSS significantly positively predicted perceived social justice ($\beta = 0.09$, $t = 3.13$, $p < 0.01$), and perceived social justice significantly positively predicted political participation ($\beta = 0.21$, $t = 7.69$, $p < 0.001$), suggesting that perceived social justice played a partial mediating role in the relationship between SSS and political participation (*indirect effect* = 0.02), thereby supporting Hypothesis 2.

**Table 1. Descriptive statistics and correlations among variables ($n = 1306$).**

|  | M | SD | 1 | 2 | 3 | 4 |
|---|---|---|---|---|---|---|
| **1. SSS** | 4.58 | 1.78 | 1 |  |  |  |
| **2. Perceived Social Justice** | 4.03 | 0.67 | 0.09** | 1 |  |  |
| **3. Materialism value** | 2.80 | 0.64 | -0.06* | -0.38*** | 1 |  |
| **4. Political participation** | 2.72 | 1.05 | 0.09** | 0.23*** | -0.18*** | 1 |

* $p < 0.05$, ** $p < 0.01$, *** $p < 0.001$, the same below.

## The moderating role of materialism value

To explore whether materialism value moderated the relationship between SSS and perceived social justice, PROCESS 4.1 Model 7 was utilized in this study, with SSS as the independent variable, perceived social justice as the mediating variable, political participation as the dependent variable, and materialism value as the moderating variable. As shown in Table 2, SSS was significantly positively correlated with perceived social justice ($\beta = 0.06$, $t = 2.51$, $p < 0.05$). More notably, the coefficient of interaction between SSS and materialism value was significant ($\beta = 0.08$, $t = 3.25$, $p < 0.01$), indicating that the relationship between SSS and perceived social justice was moderated by materialism value, thereby supporting Hypothesis 3.

To strengthen the validity of the findings, a structural equation model (SEM) was constructed to test the moderated mediation model. The results demonstrated a good fit to the data ($\chi^2/df = 1.72$, $RMSEA = 0.02$, $CFI = 0.99$, $TLI = 0.97$) [41].

To further explain the specific moderating effect of materialism value on the relationship between SSS and perceived social justice, participants were divided into the low group ($Z = -1$) and the high group ($Z = 1$) based on the standard score of materialism value. The simple slope test was conducted to investigate the effect of SSS on perceived social justice at different levels of materialism value. The results of simple slope analysis are shown in Fig 2. The results showed that for individuals with low materialism value levels, the effect of SSS on perceived social justice was not significant ($\beta = -0.03$, $t = -0.71$, $p > 0.05$). However, for individuals with high materialism value levels, SSS had a significant effect on perceived social justice ($\beta = 0.13$, $t = 3.77$, $p < 0.001$).

Moreover, bias-corrected percentile bootstrap analysis showed that the impact of SSS on political participation through perceived social justice is moderated by materialism value. For individuals with low materialism value, the indirect effect value of this influence was not significant, with the indirect effect value of -0.01($Boot\ SE = 0.01$) and a 95% confidence interval of [-0.02, 0.01]. Conversely, for individuals with a high materialism value, the indirect effect value was 0.03($Boot\ SE = 0.01$) with a 95% confidence interval of [0.01, 0.05].

The moderating roles of three materialism value dimensions (success, centrality, and happiness) on the relationship between SSS and perceived social justice were also tested. We used PROCESS Model 7, controlling for gender and age in all analyses. For material success, the interaction with SSS was not significant ($\beta = 0.04$, $p = 0.081$), indicating no moderation effect. In contrast, material centrality significantly moderated the relationship: its interaction with SSS was significant ($\beta = 0.06$, $p < 0.05$), with a stronger effect in high-centrality individuals ($\beta = 0.14$, $p < 0.001$) than low-centrality individuals ($\beta = 0.02$, $p = 0.626$). Similarly, material happiness significantly moderated the relationship: its interaction with SSS was significant ($\beta = 0.05$, $p < 0.05$). High-happiness individuals showed a significant SSS–justice link ($\beta = 0.11$, $p < 0.01$), while low-happiness individuals did not ($\beta = -0.004$, $p = 0.93$). Bootstrap analyses confirmed moderated mediation for both

**Table 2. Moderated mediation analysis ($n = 1306$).**

| Input variables | Outcome variable: Social justice | | Outcome variable: Political participation | |
|---|---|---|---|---|
| | $\beta$ | $t$ | $\beta$ | $t$ |
| Gender | 0.14 | 2.76*** | -0.13 | -2.34* |
| Age | -0.03 | -1.83 | -0.01 | -0.39 |
| SSS | 0.05 | 2.03* | 0.07 | 2.50* |
| Materialism value | -0.39 | -15.13*** | | |
| SSS×Materialism value | 0.08 | 3.26** | | |
| Perceived social justice | | | 0.23 | 8.45*** |
| $R^2$ | 0.17 | | 0.06 | |
| $F$ | 53.30*** | | 21.40*** | |

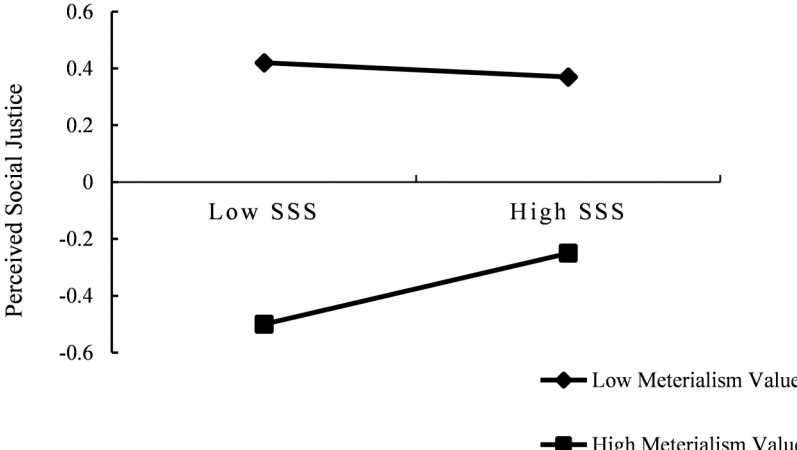

**Fig 2. The moderating effect of Materialism value on SSS and Perceived social justice.**

centrality and happiness, with indirect effects significant only in high-materialism groups (*indirect effect* $_{centrality}$ = 0.03, *95% CI* [0.01, 0.05]; *indirect effect* $_{happiness}$ = 0.03, *95% CI* [0.01, 0.05]). Thus, centrality and happiness—but not success—played a moderating role.

## Discussion

Given the inconsistent findings from previous research on the relationship between SSS and political participation, we aim to clarify this debate by examining individuals' internal psychological processes. We explored the influence of SSS on political participation and developed a moderated mediation model to articulate these dynamics.

The results showed that SSS is a significant positive predictor of political participation behavior. Moreover, perceived social justice serves as a mediator in this relationship. Meanwhile, materialism value acts as a moderator between SSS and perceived social justice, suggesting a complex interplay of individual values and socio-economic perceptions in political engagement behaviors.

### SSS, political participation, and the mediating role of perceived social justice

This study has identified SSS as a significant predictor of political participation behavior, aligning with prior findings indicating its predictive power in this regard [1]. Most of the previous research on political participation behavior primarily focuses on exploring the impact of objective SES (such as economic income) while overlooking the influence of individuals' "subjective perception" of their SES on subsequent behavior. This study's results, utilizing SSS as the independent variable, reveal its promoting effect on political participation behavior, thereby enriching the resource model.

On one hand, individuals with higher SSS tend to positively evaluate their income, education level, and other factors, which may influence their engagement in political activities [9]. Additionally, decades of social psychology research suggest that individuals usually feel, think and act with the goal of maximizing the interests of themselves or the social groups to which they belong [4]. Moreover, Social Justification Theory emphasizes that people tend to justify and defend the social systems of the group to which they belong [19]. On the other hand, the elevation of SSS also fosters a habit of political concern [21], thus motivating individuals to actively engage in politics. In summary, individuals with higher SSS demonstrate a stronger identification with their social status, possess more abundant political resources, and develop habits of political concern, thereby promoting their willingness to participate in political activities.

Our study found that perceived social justice partially mediates the relationship between SSS and political participation, through which SSS indirectly increases political participation behavior. Perceived social justice refers to individuals' subjective perceptions and value judgments regarding social fairness, including residents' evaluations of the fair distribution of important resources such as income and wealth and their psychological experiences of fairness in social contexts such as education, healthcare, and social security [19]. Motivated by self-interest, individuals with higher SSS tend to perceive society as fair, while those with lower SSS tend to perceive it as unfair. Furthermore, according to Equity Theory [43], perceived fairness affects individuals' behavioral motivations. The sense of social fairness helps individuals form positive judgments about the current social situation, thus enhancing their willingness to engage in social activities. Additionally, the 'Emotions as Social Information' model posits that the experience of fairness will trigger corresponding emotional responses in individuals, ultimately influencing their political participation behavior in terms of occurrence and degree [44]. The results of this study suggest that perceived social justice, as a positive emotional experience, is likely to serve as a connecting emotional variable between SSS and political participation, contributing to a better understanding of their relationship. Furthermore, examining the mediating role of perceived social justice not only helps to elucidate the mechanism by which SSS influences political participation but also contributes to understanding the factors that shape individual perceived social justice. This study enriches the exploration of the antecedents of perceived social justice and confirms the impact of individual psychological variables (perceived social justice) on political participation.

Given the inconsistent findings on the relationship between socioeconomic status (SES) and political participation in previous research, our study's results may hold unique significance within China's sociocultural and economic context. These factors shape how individuals perceive their socioeconomic status and, in turn, influence their actions. In China's collectivist culture, high-SSS individuals often align their personal success with national progress, viewing political participation as a civic duty to maintain social harmony [45]. This contrasts with individualistic Western societies, where high-SSS groups may prioritize self-interest over collective action [46]. Studies show that Confucian values, such as loyalty to the state and an emphasis on social order, strengthen the link between SSS and political engagement [47]. For example, urban Chinese with higher SES are more likely to endorse government policies, perceiving their participation as a contribution to national stability [48]. Additionally, China's high levels of political trust, particularly among educated and urban populations, reinforce the SSS-political participation link. Unlike in democracies where distrust in government may drive low-SSS groups to protest [15], Chinese high-SSS individuals tend to perceive the government as a legitimate arbiter of social mobility. Government-led campaigns, such as poverty alleviation programs and anti-corruption drives, are framed as evidence of systemic fairness, encouraging participation [23]. For instance, a 2020 survey found that 68% of high-SSS Chinese viewed local governance as "responsive," compared to 42% in a comparable U.S. cohort [9]. Moreover, the Chinese government strategically channels political participation through state-sanctioned platforms (e.g., online petitions, youth leagues), which are more accessible to high-SSS individuals with resources and education [32]. This contrasts with Western settings, where political engagement often requires independent mobilization [21]. These factors may collectively shape the positive relationship between SSS, perceived social justice and political participation in China. Therefore, the generalization of this study's findings should take multiple factors into account.

### The moderating role of materialism value

This study found that materialism value can weaken the positive impact of SSS on political participation behavior. SSS serves as a motivating factor for individuals to participate in political activities. The Norm Activation Model suggests that the behavioral norms generated after the power source acts on an individual mainly depend on the individual's value beliefs, with the materialism value being one such belief that forms the basis of their behavioral norms [49]. Self-Determination Theory (SDT) holds materialists play little attention to their basic needs for autonomy, competence, and relatedness and experience lower satisfaction of these needs [50]. Their focus on wealth and fame leads to a lack of concern for broader societal issues, potentially compromising their perception of justice to the whole society. For example,

when faced with the need to sacrifice their interests for the greater good of society, materialists may perceive the whole society as unfair because they focus on the satisfaction of their material interests rather than the satisfaction of internal needs. Thus, the low satisfaction of vital needs contributes to lower social justice perceptions and less likely to participate in political activities among materialists. Moreover, materialism values act as a detrimental factor that weakens the influence of SSS as a driving factor of political participation. This effect of materialism value arises from the internal mechanism of SSS triggering political participation, particularly in the initial phase of the intermediary path. This indicates that in the formation of political participation behavior, materialism value can play a role in the early stage of dynamic factors in the form of breaking factors.

Compared to previous studies, our research suggests potential psychological mechanisms underlying the influence of SSS on political participation, which provides an important theoretical foundation for improving political participation behavior. We propose that SSS positively predicts political participation through perceived social justice, indicating that individuals with high SSS perceive more sense of social justice. Consequently, in a social context characterized by a strong sense of justice, they are more willing to actively participate in political activities to advance social justice. However, materialists exhibit lower levels of sense of social justice, especially when individuals have low SSS. This is attributed to their rigid focus on self-interests and little concern for the sacred-secular boundary. The pursuit of secular (e.g., money and fame) ruled their life, and neglect the pursuit of the sacred (e.g., justice, and love), thus, they unconsciously view the secular and scared are opposed and refuse to make a trade-off between them [17]. When they perceive that their material interests are not satisfied, they will not think that it is their reasons, but tend to think that the whole society is unfair to them, and it is the society that sacrifices their interests to achieve "justice", so they are more reluctant to participate in political activities.

Notably, two of the three dimensions of materialism—centrality and happiness—play distinct roles in moderating the SSS–justice link. Material centrality, which prioritizes possessions as the core purpose of life, amplifies the belief that SSS reflects personal merit [36]. Individuals with high material centrality are more likely to interpret their socioeconomic status as evidence of a fair system, a pattern particularly observed among urban Chinese youth [32]. Similarly, material happiness, which associates well-being with access to material resources, reinforces perceptions of fairness among high-SSS individuals [35]. In contrast, material success, which reflects the belief that personal achievement is closely tied to the accumulation of wealth and possessions, does not significantly moderate the SSS–justice link. This may be because material success emphasizes individual accomplishment rather than social comparison [51]. As a result, individuals with strong material success values may less likely to attribute low SSS to social injustice and do not exhibit an enhanced perception of justice at high SSS levels, which may explain the absence of a moderating effect.

## Practical implications

The findings of this study have significant implications for policymakers and educators. First, the mediating role of perceived social justice suggests that interventions aimed at enhancing fairness in resource distribution—such as transparent governance and equitable access to education—could mitigate political apathy among low-SSS groups, thereby promoting greater engagement. For example, grassroots campaigns emphasizing social equity (e.g., anti-corruption initiatives) may foster trust in institutions and encourage participation [23]. Second, the moderating effect of materialism highlights a dual challenge in China's rapidly modernizing society, where economic growth raises material aspirations, yet excessive materialism may undermine collective values essential for civic engagement [32]. Educational programs promoting intrinsic values (e.g., community service, ethical leadership) could counterbalance materialistic tendencies, particularly among youth. For example, integrating Confucian principles of social harmony into curricula might strengthen the link between SSS and prosocial political behaviors [52]. Third, the study's focus on university students—a demographic central to China's future governance—underscores the need for campus initiatives that bridge socioeconomic divides. Such

initiatives could include policies like financial aid for low-SSS students or platforms for inclusive political discourse (e.g., student councils), both of which would help reduce perceived inequities and empower marginalized groups to engage more actively [47].

## Limitations and future research

First, the measure of political participation used in this study mostly reflects moderate forms of engagement, aimed at maintaining the existing system rather than instigating significant change. Previous research often examined more confrontational forms of engagement, such as collective protests [17], which may explain some differences between our findings and those of other studies. Furthermore, this study did not thoroughly analyze how political participation structures vary among individuals with different socioeconomic statuses. Future research should investigate whether forms of political participation differ by socioeconomic status and the underlying process mechanisms.

Second, although our results support the mediating role of perceived social justice, alternative mechanisms warrant consideration. On one hand, complacency among high-SSS individuals, as noted in prior studies [15], could coexist with the observed mediation. For instance, high-SSS individuals may engage in "ritualistic" participation (e.g., voting) to maintain status quo rather than drive change, reflecting system justification rather than genuine activism. On the other hand, nonlinear relationships may exist. For example, at extreme SSS levels, individuals might disengage due to overconfidence in systemic fairness or resource saturation. Future studies should test quadratic or moderated nonlinear models to capture these nuances.

Third, although our study integrated models related to political participation behavior, it focused solely on individual psychological variables and overlooked environmental influences. Future research should consider environmental factors, such as online environments, to provide a more comprehensive view.

Fourth, this study is a self-reported questionnaire study, which may be subject to common method bias [53]. Although the study controlled for this potential issue through procedures such as eliminating item ambiguity, anonymous testing, and conducting Harman's single-factor test, method bias may still affect the results. Future research could employ more diverse measurement methods, such as longitudinal designs, behavioral indicators, and experimental designs, to examine the relationships between variables and eliminate the impact of method bias on the results.

Finally, our homogeneous sample (Chinese university students) limits generalizability. In more diverse populations (e.g., in other countries or cultures, or among older adult groups), SSS might correlate differently with political participation due to varying access to resources or exposure to state narratives [54]. Cross-cultural replications are critical: in individualistic societies, materialism might weaken the SSS–justice link by prioritizing self-interest over collective fairness [32]. Comparative studies could disentangle cultural versus structural influences. Future research could include participants from different socioeconomic and cultural backgrounds.

## Conclusion

We explored the relationship among SSS, perceived social justice, materialism value, and political participation. Our findings indicated that political participation is influenced not only by SSS, but also by internal factors such as materialism value and perceived social justice. Notably, perceived social justice plays an intermediary role between SSS and political participation. Materialism value moderates the initial segment of the mediating pathway. Specifically, under conditions of high materialism value, SSS predicts perceived social justice. whereas under conditions of low materialism value, SSS does not show a significant predictive effect.

These findings suggest that models of political behavior must integrate aspects of individual materialism and perceived social justice to fully capture the nuances of how socioeconomic status influences civic engagement. Such an integrative approach could provide a more holistic understanding of political participation and offer a more comprehensive framework for future research in political psychology.

## Supporting information

**S1 File. Scale Development.** Development of the Political Participation Behavior Scale.
(DOCX)

**S2 Data. The original data.**
(XLSX)

**S3 Checklist. Inclusivity in global research questionnaire.**
(DOCX)

## Author contributions

**Conceptualization:** Zhirui Zhao.

**Data curation:** Qi Zhao.

**Formal analysis:** Zhirui Zhao.

**Methodology:** Zhirui Zhao.

**Writing – original draft:** Zhirui Zhao.

**Writing – review & editing:** Su Tao, Wenchong Du.

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
