## [Decision Letter · Decision Letter 0]

6 Feb 2025

PONE-D-24-46923Materialists Perceive their high socioeconomic status as justice: Leading to increased political participationPLOS ONE

Dear Dr. Tao,

Thank you for submitting your manuscript to PLOS ONE. After careful consideration, we feel that it has merit but does not fully meet PLOS ONE’s publication criteria as it currently stands. Therefore, we invite you to submit a revised version of the manuscript that addresses the points raised during the review process.

We look forward to receiving your revised manuscript.

Kind regards,

Enrico Ivaldi

Academic Editor

PLOS ONE

Journal Requirements:

3. Please include a complete copy of PLOS’ questionnaire on inclusivity in global research in your revised manuscript. Our policy for research in this area aims to improve transparency in the reporting of research performed outside of researchers’ own country or community. The policy applies to researchers who have travelled to a different country to conduct research, research with Indigenous populations or their lands, and research on cultural artefacts. The questionnaire can also be requested at the journal’s discretion for any other submissions, even if these conditions are not met.  Please find more information on the policy and a link to download a blank copy of the questionnaire here: https://journals.plos.org/plosone/s/best-practices-in-research-reporting. Please upload a completed version of your questionnaire as Supporting Information when you resubmit your manuscript.

“This research was funded by Humanities and Social Science Projects of the Ministry of Education of China (19YJC710066).”

5. In the online submission form, you indicated that “The original data of this paper can be obtained by contacting the corresponding author.”

Reviewers' comments:

Reviewer's Responses to Questions

**Comments to the Author**

1. Is the manuscript technically sound, and do the data support the conclusions?

Reviewer #1: Partly

Reviewer #2: Partly

2. Has the statistical analysis been performed appropriately and rigorously? 

Reviewer #1: Yes

Reviewer #2: Yes

3. Have the authors made all data underlying the findings in their manuscript fully available?

Reviewer #1: No

Reviewer #2: Yes

4. Is the manuscript presented in an intelligible fashion and written in standard English?

Reviewer #1: Yes

Reviewer #2: Yes

5. Review Comments to the Author

Reviewer #1: I am grateful for the opportunity to review this manuscript. It presents an interesting research topic. The authors have employed a quantitative research method to examine the relationships among the variables, which is refreshing. However, there are several concerns that need to be addressed:

Q1: The correlations presented in the abstract should reflect the correlations among all variables in the study, including the analysis of materialism value.

Q2: The introduction and literature review would benefit from more literature evidence. Generally, statements that are not the authors' own opinions require citations. The lack of citations is a significant issue in this manuscript, a large lack of literature citations can lead reviewers to question the necessity and rigor of the study.

Q3: There are many instances in the introduction and literature review where it states "numerous studies indicate...". There should be at least three references to support this claim.

Q4: The proposed moderated mediation model (Fig. 1) should clarify whether the moderating variable has a positive or negative moderating effect.

Q5: Please elaborate on the sample selection process or reason in the research methods, as this will shed light on sample representativeness.

Q6: What is the basis for the minimum sample size, and how to control the bias in advance (please note that Harman’s single-factor test is a post-hoc bias test)? For further learning on bias control, you may refer to the following literature:

Podsakoff, P. M., MacKenzie, S. B., Lee, J. Y., & Podsakoff, N. P. (2003). Common method biases in behavioral research: a critical review of the literature and recommended remedies. Journal of applied psychology, 88(5), 879. https://doi.org/10.1037/0021-9010.88.5.879

Q7: What platform was used for online questionnaire collection? Please provide a valid link to that platform.

Q8: More demographic background information about the participants should be provided, such as their grade and family location, as these background variables may affect the stability of the research results.

Q9: I noticed that the other three measurement tools used by the authors are Likert scales. Is the SSS scale also a Likert scale? If not, how are the scores converted of SSS scale? Additionally, please provide reliability and validity analysis for the SSS scale.

Q10: As far as I know, developing a self-constructed questionnaire or scale requires substantial preliminary work, such as: how to generate an item pool? Is item analysis and exploratory factor analysis conducted? To ensure the reliability and validity of the tool. Although this study uses CFA, it lacks the necessary steps for a self-constructed questionnaire, which raises doubts about the measurement tool.

Q11: Please provide the criteria or reference ranges for all indicators in the research methods and results, such as Cronbach's α, model fit indices, and the critical threshold of 40%.

Q12: Please detail the data analysis strategy used in this study.

Q13: The descriptive statistics and correlation analysis of the research results should include the moderating variable, materialism value.

Q14: The discussion section would benefit from more literature support, and incorporating discussions on the Chinese cultural context could further explain the validity of the research results.

Q15: Please clarify the practical value of this research.

In summary, this manuscript requires more effort in literature citations to meet the standards of academic writing.

Reviewer #2: The study provides valuable insights into the interplay of socioeconomic perceptions, justice beliefs, and material values in shaping political participation. Below, I provide detailed feedback and pose questions for the authors to address in their revision.

General Feedback:

The paper presents a well-structured investigation into the relationship between subjective socioeconomic status (SSS) and political participation. The findings contribute meaningfully to political psychology and behavioral economics by integrating psychological mechanisms such as perceived social justice and materialism. The use of established scales and robust statistical techniques enhances the credibility of the study. However, certain areas require further clarification and elaboration to strengthen the manuscript.

Major Points for Revision:

• The authors acknowledge contradictory findings in the literature regarding SSS and political participation. While this study finds a positive relationship, some studies suggest a negative or null effect. Could the authors provide a more detailed discussion of why their findings differ from others? Are contextual factors (e.g., China's political environment, university culture) potentially influencing this relationship?

• The study suggests that individuals with higher SSS perceive greater social justice, which in turn fosters political participation. However, previous research indicates that individuals with higher SSS might disengage from political participation due to a sense of complacency. Can the authors explore alternative explanations for the mediation effect? Could there be a nonlinear or curvilinear relationship that was not tested? However, it is possible that relationships are more complex. The authors did not test for any non-linear relationships and assume linear relationships between variables. For example, the relationship between SSS and political participation may not be consistently positive. It is possible that at very high levels of SSS, individuals may become complacent or disengaged from political activities. Similarly, there might be a threshold effect where only after a certain level of perceived social justice has been reached do individuals become more politically active. This might result in a curvilinear relationship, which the study did not explore.

• The findings indicate that materialism strengthens the relationship between SSS and perceived social justice. However, materialism is a multidimensional construct with facets such as success, centrality, and happiness. Although the study does not explore whether all three dimensions of materialism have different moderating effects, it is plausible that they do. Future research could explore these differences to develop a more comprehensive understanding of how materialism influences the relationship between SSS, perceived social justice, and political participation. Do all dimensions of materialism exert the same moderating effect? If not, how do they differ?

• The study relies on PROCESS models which are well-established but have limitations. Could the authors consider alternative methods such as Structural Equation Modeling (SEM) or Bayesian Mediation Analysis? By using alternative methods like SEM or Bayesian analysis, the authors could gain a more nuanced and comprehensive understanding of the complex relationships among SSS, perceived social justice, materialism and political participation. These methods would provide a more robust assessment of the model's validity and offer the possibility of exploring more complex relationships. A comparison of methods could add robustness to the findings.

Additional Points for Consideration:

• Would a more heterogeneous sample yield different findings?

• Given China's unique political and economic landscape, to what extent do the findings generalize to other cultural settings?

Overall, this study offers valuable contributions to the understanding of political participation among Chinese college students. The use of a moderated mediation model is commendable, but further clarifications, additional robustness checks, and theoretical discussions would enhance the manuscript. I encourage the authors to address these points in their revision.

6. PLOS authors have the option to publish the peer review history of their article (what does this mean? ). If published, this will include your full peer review and any attached files.

**Do you want your identity to be public for this peer review?** For information about this choice, including consent withdrawal, please see our Privacy Policy .

Reviewer #1: No

Reviewer #2: No

---

## [Author Response · Author response to Decision Letter 0]

20 Mar 2025

Dear reviewers,

We sincerely appreciate your valuable suggestions, which are very helpful for us to modify the manuscript. Our point-by-point responses are provided in The file “Response to Reviewers”.

If you have any further comments or suggestions, please feel free to share them, we would be happy to address them.

---

## [Decision Letter · Decision Letter 1]

25 Apr 2025

PONE-D-24-46923R1Materialists Perceive their high socioeconomic status as justice: Leading to increased political participationPLOS ONE

Dear Dr. Tao,

Thank you for submitting your manuscript to PLOS ONE. After careful consideration, we feel that it has merit but does not fully meet PLOS ONE’s publication criteria as it currently stands. Therefore, we invite you to submit a revised version of the manuscript that addresses the points raised during the review process.

**The authors are requested to revise the title of the manuscript by removing the term 'lead to,' as it may imply a causal relationship that is not supported by the cross-sectional study design.**

We look forward to receiving your revised manuscript.

Kind regards,

Enrico Ivaldi

Academic Editor

PLOS ONE

**Journal Requirements:**

**Additional Editor Comments:**

The authors are requested to revise the title of the manuscript by removing the term 'lead to,' as it may imply a causal relationship that is not supported by the cross-sectional study design.

Reviewers' comments:

Reviewer's Responses to Questions

**Comments to the Author**

1. If the authors have adequately addressed your comments raised in a previous round of review and you feel that this manuscript is now acceptable for publication, you may indicate that here to bypass the “Comments to the Author” section, enter your conflict of interest statement in the “Confidential to Editor” section, and submit your "Accept" recommendation.

Reviewer #1: All comments have been addressed

Reviewer #2: All comments have been addressed

2. Is the manuscript technically sound, and do the data support the conclusions?

Reviewer #1: Yes

Reviewer #2: Yes

3. Has the statistical analysis been performed appropriately and rigorously? 

Reviewer #1: Yes

Reviewer #2: Yes

4. Have the authors made all data underlying the findings in their manuscript fully available?

Reviewer #1: Yes

Reviewer #2: Yes

5. Is the manuscript presented in an intelligible fashion and written in standard English?

Reviewer #1: Yes

Reviewer #2: Yes

6. Review Comments to the Author

**Reviewer #1:**  I would like to express my appreciation for the authors’ efforts in revising the manuscript. The quality of the revised version has been significantly improved, and the authors have adequately addressed my concerns. It is evident from the revised manuscript and the response letter that the research team has dedicated considerable effort to this work. However, given that your study is cross-sectional in design, I recommend revising the title to remove the term “lead to,” which may imply a causal relationship. Once this modification is made, I am prepared to recommend the manuscript for publication.

**Reviewer #2: ** (No Response)

7. PLOS authors have the option to publish the peer review history of their article (what does this mean? ). If published, this will include your full peer review and any attached files.

**Do you want your identity to be public for this peer review?** For information about this choice, including consent withdrawal, please see our Privacy Policy .

Reviewer #1: **Yes: ** Jun Li

Reviewer #2: No

---

## [Author Response · Author response to Decision Letter 1]

26 Apr 2025

Thank you very much to the editor and reviewers for your valuable comments.

We agree with your concern that the phrase “lead to” may suggest a causal relationship. After careful discussion, we have revised the title to “Materialists Perceive Their High Socioeconomic Status as Justice: Associations with Increased Political Participation” to better reflect a correlational rather than a causal relationship.

We sincerely hope that our revisions address your concerns. Should you have any further questions, please do not hesitate to contact us.

---

## [Editor Report · Decision Letter 2]

29 Apr 2025

Materialists Perceive Their High Socioeconomic Status as Justice: Associations with Increased Political Participation

PONE-D-24-46923R2

Dear Dr. Tao,

We’re pleased to inform you that your manuscript has been judged scientifically suitable for publication and will be formally accepted for publication once it meets all outstanding technical requirements.

Kind regards,

Enrico Ivaldi

Academic Editor

PLOS ONE
---

## [Editor Report · Acceptance letter]

PONE-D-24-46923R2

PLOS ONE

Dear Dr. Tao,

I'm pleased to inform you that your manuscript has been deemed suitable for publication in PLOS ONE. Congratulations! Your manuscript is now being handed over to our production team.

Kind regards,

on behalf of

Prof. Enrico Ivaldi

Academic Editor

PLOS ONE